# Arabinoxylan Concentrate from Wheat as a Functional Food Ingredient to Improve Glucose Homeostasis

**DOI:** 10.3390/nu17091561

**Published:** 2025-04-30

**Authors:** Knud Erik Bach Knudsen, Helle Nygaard Lærke, Mette Skou Hedemann, Kirstine Lykke Nielsen, Mirosław Marek Kasprzak, Per Bendix Jeppesen, Merete Lindberg Hartvigsen, Kjeld Hermansen

**Affiliations:** 1Department of Animal and Veterinary Sciences, Aarhus University, Blichers Allé 20, 8830 Tjele, Denmark; helle.laerke@anivet.au.dk (H.N.L.); mette.hedemann@anivet.au.dk (M.S.H.); klyn@forens.au.dk (K.L.N.); m.kasprzak@urk.edu.pl (M.M.K.); 2Department of Forensic Medicine, Arhus University, Palle Juul-Jensens Boulevard 99, 8200 Aarhus, Denmark; 3Department of Animal Product Technology, University of Agriculture, Balicka 122, 30-149 Cracow, Poland; 4Department of Clinical Medicine, Aarhus University Hospital, Palle Juul-Jensens Boulevard 165, 8200 Aarhus, Denmark; per.bendix.jeppesen@clin.au.dk (P.B.J.); mehar@arlafoods.com (M.L.H.); kjeld.hermansen@clin.au.dk (K.H.); 5Arla Food Ingredient Group P/S, 8260 Viby, Denmark; 6Department of Endocrinology and Internal Medicine, Aarhus University Hospital, Palle Juul-Jensens Boulevard 99, 8200 Aarhus, Denmark

**Keywords:** arabinoxylan, dietary fiber, glucose, insulin, homeostasis, humans, animals

## Abstract

*Background:* Arabinoxylan (AX) concentrates from wheat can be produced from co-products from the starch and gluten industries. These fractions are rich in AX, have high solubility, can be incorporated into bread and breakfast cereals, and have the potential to enhance functional and nutritional effects beyond what is possible with cereal species. *Methods:* The aim of this review is to summarize the available literature on AX concentrates from wheat in terms of production, incorporation in breads, and influence on glucose homeostasis in human subjects and animals. *Results:* Breads enriched in AX fiber have been found to increase the viscosity of digesta from the small intestine but with no effect on the digestibility of starch. In the large intestine, AX is, to a large extent, degraded, producing short-chain fatty acids. Across acute human and animal studies, the intake of AX concentrates has been shown to reduce the rate and extent of glucose absorption and insulin responses in a dose-dependent fashion. No general influence of AX on incretins has been identified, and the role of AX-enriched diets in appetite sensation is unclear. Medium- and long-term human and animal intervention studies demonstrated improved glucose homeostasis (fructosamine and glycated hemoglobin A1c) during the consumption of AX-enriched diets compared to placebo. Although AX concentrates across studies improved glucose homeostasis, a confounding factor to be further investigated is to what extent protein being present in relatively high concentrations in some of the produced AX-rich wheat fractions, partly or fully, is responsible for the observed homeostatic effects.

## 1. Introduction

The types of diets consumed in Western societies are rich in readily digestible carbohydrates that cause rapid postprandial elevations in glucose and insulin [1,2]. Long-term exposure to such diets is often associated with a gradual increase in body weight, which in the long term can lead to obesity, metabolic syndrome, cardiovascular disease, and type 2 diabetes (T2DM) [3]. A factor known to influence the rate of nutrient absorption is the level of soluble dietary fiber (DF). Soluble DF can form viscous solutions or gels in the stomach, which delay gastric emptying and physically slow the digestion and absorption of macronutrients in the small intestine [4,5]. This is positively correlated with lower postprandial glucose and insulin responses [4,5]. The release of insulin is also influenced by incretin hormones, glucose-dependent insulinotropic polypeptide (GIP), and glucagon-like peptide 1 (GLP-1), which are gut peptides secreted by enteroendocrine K and L cells in the upper and lower small intestine, respectively. As with insulin, incretin hormones are secreted after nutrient intake and, together with hyperglycemia, stimulate insulin secretion [6]. The rate of glucose absorption is known to influence the release of incretin [6], and soluble DF may, in this respect, be a food-modulating factor [7,8]. Short-chain fatty acids (SCFAs) produced by the microbiota in the large intestine, and propionate in particular, stimulate GLP-1 secretion [9,10]. However, while the GLP-1 secretion caused by nutrients absorbed in the small intestine directly affects insulin secretion, the effect of SCFAs on the GLP-1 secretion from the L cells in the large intestine is functionally different and acts as an energy sensor [11,12].

Cereals in the form of breakfast cereals and breads (white wheat bread, brown bread, whole grain breads, bran-enriched breads, etc.) are the main contributors to the intake of DF in Denmark, accounting for ~60% of the intake [13]. Most of the wheat breads consumed are produced from refined wheat flour and with a DF content that is approximately one-fourth of the DF in whole grain wheat [14] and even less than that of whole grain rye, the most commonly consumed bread in Denmark [14]. Although there has been a trend towards increased consumption of whole grain products due to taste, texture, and health recommendations [15,16], the use of the outer bran layers reduces the volume of the bread and is associated with a more bitter taste, which limits the use of whole grain flour in bread production [17]. An alternative approach to the use of whole grain flour can be to increase the DF content by using concentrated sources of DF. For example, arabinoxylan (AX), which can be obtained from all cereals [18], and mixed-linked (1-3)(1-4)-β-D-glucan (β-glucan) can be isolated from the cell walls of oats and barley [19]. Beta-glucan has been approved by the Food and Drug Administration in the US [20] and the European Food Safety Authorities [21] for health claims. AX from wheat and rye may hold the same potential as β-glucan to regulate the digestion and absorption of nutrients from the small intestine.

The main purpose of this review is to summarize current literature on the impact of AX concentrates from wheat on digestion, absorption, and glucose homeostasis. The core data for the review were derived from the Danish Strategic Research funded project “Biofunctional Carbohydrates”, an interdisciplinary project using in vitro and in vivo models (pigs, Zucker diabetic fatty rats, and humans) combined with a literature search in Web of Science (WoS). The key words for the WoS search were “arabinoxylan” and “concentrate”, and “glucose” or “arabinoxylan” and “insulin” and “GLP-1” or “arabinoxylan” and “metabolomics” or “arabinoxylan” and “satiety”.

## 2. Arabinoxylan as an Ingredient and Its Incorporation in Breads

The main DF components present in cereals are AX, β-glucan, cellulose, and lignin. AX and β-glucan are present as both soluble and insoluble components, whereas cellulose and lignin are insoluble. AX is the main cell wall component of the endosperm, aleurone, and pericarp testa layers of all cereals [18], but wheat and rye are of special interest, as the two cereals are commonly used to produce breads. AX consists of a linear backbone of β-(1,4)-linked d-xylose residues commonly decorated with α-l-arabinose substitutions at O-2 and/or O-3 and ferulic acid (Figure 1) [22,23]. The AX structure is reasonably similar in wheat and rye, but the relative solubility depends on the location in the grain, with the highest solubility in the endosperm and lowest in the aleurone and pericarp/testa tissues; the latter tissues make up the bran fraction (Figure 1). The functional properties of AX are influenced by the extent of the substitution and distribution of substituents along the xylan backbone, which is reflected in different ratios of arabinose to xylose (A:X) [22,23]. Furthermore, the properties of DF in the liquid and solid phases are affected by the molecular weight (M_w_) of the polymers, their concentrations, non-covalent interactions (e.g., hydrogen linkages), and covalent bonds (e.g., diferulic bridges) with other cell components and nutrients [24,25].

Dry or wet separation technologies can be used to produce DF-rich fractions with distinct properties [27]. AX concentrate can, for example, be produced from co-products from wheat starch and/or gluten production [27]. Table 1 illustrates the composition of three AX-rich concentrates produced in sufficient quantities to be used in human and pig studies from the co-product after wheat starch and gluten production. Fraction I was collected onto a sieve (75 μm), washed thoroughly with water, and spray-dried to a powder [28]. Fraction II was extracted from the process water after wheat starch production and further processed on a pilot plant involving enzymatic treatment, yeast fermentation, ultrafiltration, and spray-drying [29]. Fraction III was produced by concentrating the soluble fraction after starch and gluten extraction, treating it with heat and enzymes, precipitating the soluble DF with ethanol (1:3 *v*/*v*), filtering, and finally spray-drying [30]. The different processing procedures give rise to differences in composition. The concentration of fat was low in all the fractions, but there were large variations in the concentrations of non-starch polysaccharides (NSPs), AX, and protein, and the solubility of AX. For Fraction I, the A:X ratio was of the same order as that found for whole grain wheat, white flour, and bran [14], whereas it was higher for Fraction II; in particular, Fraction III showed a higher degree of branching of the AX in these fractions. The average M_w_ of Fraction II was reported to be 20–40 kDa [31], and for Fraction III, it was 602 kDa [32]. The M_w_ of Fraction III was of the same order as that found for soluble AX in wheat [33]. For comparison, the M_w_ of β-glucan was higher, at 1978 kDa [32], similar to what was reported by Åman et al. for oat β-glucan [34].

An AX concentrate with a purity higher than 80% was produced by Luo et al. [35] by isolating the supernatant from the wastewater after wheat gluten production, sequentially concentrating the soluble AX in the liquid phase, filtrating it, deproteinizing it, removing ethanol via rotary evaporation, and finally, vacuum freeze-drying to obtain purified AX.

**Table 1 nutrients-17-01561-t001:** Composition (g/kg dry matter) of whole grain wheat and three AX-rich fractions.

	Whole Grain Wheat	AX-Rich Fraction I	AX-Rich Fraction II	AX-Rich Fraction III
Wheat processing		Starch and gluten	Starch	Starch and gluten
Ash	18	nm	13	72
Protein	138	101	170	397
Fat	29	Trace	4	2
Starch	649	156	nm	40
Total NDC	131	nm	nm	464
Total NSP	115	742	nm	312
Soluble:total NSP	0.26	0.62	nm	0.91
AX	75	668	566	234
Arabinose:xylose	0.62	0.66	0.80	0.94
AX-oligosaccharides	nm	nm	nm	95
Reference	#	[28]	[29]	[30]

AX, arabinoxylan; NDC, non-digestible carbohydrate; NSP, non-starch polysaccharides; nm, not measured. # The results for whole grain wheat are based on data published in [36] and complemented with other in-house samples (N = 16).

The AX-rich fractions have been used as ingredients in breads for nutritional studies. An example is provided in Table 2 showing the composition of an AX-rich bread (AXB) compared with low-DF white flour bread (WWB), a β-glucan-enriched bread (BGB), and two commercial whole grain rye breads without and with kernels (WRB and WRBK); for details on the ingredient compositions of these breads, see [30,37,38]. It is worth noting that the total content of AX is about the same in AXB as in the whole grain rye breads (WRB and WRBK), but the soluble AX is almost 80% higher due to the high solubility of the concentrated AX ingredient. After bread making, there was a two-fold reduction in the M_w_ of the AX in AXB to a level lower than that of the AX in WRBK but higher than for WRB [32]. For comparison, the M_w_ of the β-glucan in BGB was reduced eight-fold [32].

There is limited information on how the AX concentrates influence the sensory quality of the bread, but in a study where breads containing two levels of AX-rich fiber were evaluated by a panel of 30 assessors and compared to a control bread, there were no significant differences in the mean scores between the control bread and the breads containing AX-rich fiber regarding flavor, color, texture, and overall quality [28].

## 3. Functional Properties of AX in the Gastrointestinal Tract and Influence on Digestion

AX in concentrated form has been found to give rise to a higher luminal viscosity in ileal digesta than concentrated β-glucan and when AX is present in whole grain rye bread (Table 3) [30,32]. A positive correlation between the concentration of NSP in the liquid phase of ileal digesta and the log-transformed viscosity (r = 0.59, *p* = 0.0017) was found [30]. The higher ileal viscosity with the AX concentrate, however, had no influence on the ileal digestibility of starch. This contrasted with what was found for the β-glucan-rich bread (BGB) and the bread with intact kernels (WRBK). For the latter diets, the lower digestibility of starch is most likely due to two different mechanisms; the intact kernels present in the WRBK diet present a physical barrier and impair the accessibility of the starch, while the high M_w_ of the β-glucan in the BGB diet can create a highly viscous environment in the duodenum that decreases the access of starch-degrading enzymes to the substrate by limiting the water available for starch hydration as proposed and discussed by Lazaridou and Biliaderis [39]. The results are striking in light of a much higher degradation and depolymerization of β-glucan than of AX in the small intestine [30,32].

Adding 15 g of AX-rich fiber per day to a control diet containing 34 g of total fiber per day increased bowel movements and caused a significantly higher fecal output than the control diet (284 ± 36 and 223 ± 30 g/d, respectively, *p* = 0.05) [40]. The polysaccharide content of the feces was also higher after the AX diet (15.1 ± 2.9 and 17.2 ± 2.8 g/day, control and AX diet, respectively, *p* = 0.02). If the extra output of polysaccharides in feces is compared to the extra intake of DF with the AX diet, it is clear that the majority of the AX is degraded in the gastrointestinal tract, giving rise to enhanced SCFA production. In pigs, it is known that the main site for the degradation of soluble AX is the cecum and proximal colon [41,42], where the highest SCFA concentration is also found when a diet enriched in AX-rich fiber is compared to the control [43]. An acute study with semolina porridge fed alone or with added AX concentrate, rye kernels, or a combination of AX concentrate and rye kernels (1:1) further demonstrated a significant and rapid rise in breast hydrogen, a marker for fermentation in the large intestine, after 240 min and significant rises in the plasma concentrations of acetate and butyrate but not propionate for the AX-enriched diet and the combined AX and rye kernel diet after 360 min [38]. The feeding of AX concentrate to pigs also seems to influence the bacterial populations as indicated by a significantly higher abundance of some species from the Prevotella cluster (*Prevotella intermedia*, *P. disiens*, and *P. ruminicola*) from the Bacteroidetes phylum and from Clostridial clusters IV (*Faecalibacterium prausnitzii*) and *XIVa* (*Ruminococcus obeum* and *Blautia producta*) in the AX-rich diet for both mucosa and digesta fractions [43]. A study with type 2 diabetic rats also showed that supplementation with arabinoxylan isolated from the seeds of *P. asiatica* L., a non-cereal source, promoted the growth of fiber-degrading bacteria and increased SCFAs [44]. To the authors’ knowledge, there are no studies on the influence of AX concentrate from wheat on the microbiota composition in humans, but it can be noted that AX from psyllium husk (not cereals) provided in increasing doses of 5 g/d, 10 g/d, and 15 g/d over a three-week period influenced the microbial composition; for AX, there was a dose–response increase with fiber in the genus *Roseburia* and species *Bacteroides xylanisolvens*—two taxa known to ferment fibers in the colon [45]. A human study with overweight human individuals with indices of metabolic syndrome provided arabinoxylan oligosaccharides produced from wheat demonstrated a bifidogenic effect and a boost in the proportion of *Prevotella* species [46]. Moreover, a diet enriched with AX and resistant starch resulted in significant changes in the microbiota community with an increased proportion of *Bifidobacterium* and decreased proportions of certain bacterial genera associated with dysbiotic intestinal communities for the diet rich in AX and resistant starch [47]. The total SCFA, acetate, and butyrate concentrations were also higher when a diet rich in AX and resistant starch was fed.

## 4. Influence of AX on Glucose Absorption and Regulation

### 4.1. Acute Studies

The rate of glucose absorption is regulated by the rate of gastric emptying and the rate of the digestion and absorption of starch in the small intestine (Figure 2). The physiochemical properties of DF mean that, depending on its chemical and structural composition, it can delay gastric emptying, interfere with the hydrolysis processes in the lumen of the small intestine, and slow the movement of hydrolysis products from the lumen through the mucus layer and to enterocytes for absorption [5,48]. These aspects have been well described in studies with isolated soluble DF sources such as guar gum, pectin, and β-glucan [4,5,48]. AX concentrates have similar properties to those of DF sources and, when incorporated into bread, gave rise to higher luminal viscosity in the small intestine than when present naturally in whole grain rye (Table 3). Although concentrated AX had no effect on the digestibility of starch in the small intestine, multiple acute human and animal studies have shown a reduced rate and extent of glucose absorption (Table 4 and Table 5). The acute human studies listed in the tables were essentially performed in a similar manner to when evaluating the glycemic index (GI) of foods. The GI is a system for the exchange of carbohydrates between foods and is defined as the area under the blood glucose curve (here denoted as the incremental area under the curve (iAUC)) following the ingestion of a test food, expressed as a percentage of the corresponding area following an equivalent load of carbohydrate from a reference, in this case, white wheat bread [49].

**Table 4 nutrients-17-01561-t004:** Acute human and animal studies on the effects of arabinoxylan on glucose homeostasis.

AX Source	AX Dose and Control	Species	Parameters Studied	Observed Effect	Reference
AX extracted from wheat co-products after starch and gluten extraction	6 or 12 g of AX provided in breads and compared with white wheat bread as a control	Human normoglycemic subjects	Plasma glucose, insulin, and iAUC	Dose-dependent effects of AX on iAUC glucose and iAUC insulin in normoglycemic subjects	[28]
AX concentrate from the soluble fraction after wheat starch and gluten extraction	7 g of AX provided in bread and compared to white wheat bread, whole grain rye bread with kernels, and β-glucan-rich bread	Human subjects with metabolic syndrome	Plasma glucose, insulin, GLP-1, GIP, ghrelin, iAUC, and appetite score	AX reduced peak glucose but neither the initial glycemic response nor insulin response. AX increased satiety compared to the white wheat bread but did not result in a significant difference in subsequent ad libitum energy intake after 270 min.	[38]
AX concentrate from the soluble fraction after wheat starch and gluten extraction	3.5 g of AX provided alone or in combination with whole rye kernels (4.4 g AX) compared to semolina porridge	Human subjects with metabolic syndrome	Plasma glucose, insulin, GLP-1, ghrelin, iAUC, breast hydrogen, plasma short-chain fatty acids, and appetite score	AX combined with rye kernels reduced acute glucose and insulin responses and feelings of hunger compared with the control meal. AX alone and in combination with rye kernels increased butyrate and acetate concentrations after 6 h compared to control but with no differences in the second meal response for glucose, insulin, free fatty acids, glucagon-like peptide-1, or ghrelin	[50]
AX-enriched white bread flour	3.2 g of AX-enriched bread compared with white wheat bread	Human normoglycemic subjects	Plasma glucose	The 30 min peak plasma glucose concentration after AX-enriched meals was significantly lower than that for control white wheat bread	[51]
AX concentrate from the soluble fraction after wheat starch and gluten extraction	AX provided in bread and compared to white wheat bread, whole grain rye bread, whole grain rye bread with kernels, and β-glucan-rich bread	Portal vein-catheterized pigs	Plasma glucose, insulin, glucose absorption, GLP-1, and GIP	Net portal glucose absorption was reduced in pigs fed the AX bread at 60 min and insulin secretion was lowered at 30 min with AX bread and whole grain rye bread compared to white wheat bread	[37]

AX, arabinoxylan; iAUC, incremental area under the curve; GLP-1, glucagon-like peptide-1; GIP, glucose-dependent insulinotropic peptide

**Table 5 nutrients-17-01561-t005:** Main effects of the inclusion of arabinoxylan relative to control in diets in acute human and animal studies on parameters related to glucose homeostasis.

	AX	iAUC		Concentration					
	g or % of DM	Glucose	Insulin	Glucose	Insulin	GLP-1	GIP	Ghrelin	Ref.
AX extracted from wheat co-products after starch and gluten extraction	6 g	↓	↓	↓	→				[28]
	12 g	↓↓	↓	↓↓	↓				
AX concentrate from the soluble fraction after wheat starch and gluten extraction	7 g	→	→	↓	↑	→	→	→	[38]
AX concentrate from the soluble fraction after wheat starch and gluten extraction	3.5 g	↓	→	→	→	→		→	[50]
	4.4 g	→	↓	↓	↓	→		→	
AX-enriched white bread flour	3.2 g			↓					[51]
AX concentrate from the soluble fraction after wheat starch and gluten extraction	7.8% vs. 1.7% ^1^	→	→	↓ ^2^	↓ ^2^	→	→		[37]

iAUC, incremental area under the curve; AX, arabinoxylan; GLP-1, glucagon-like peptide-1; GIP, glucose-dependent insulinotropic peptide. ↓ significantly reduced compared to control (*p* < 0.05); ↓↓ significantly reduced compared to control (*p* < 0.01); → not significantly different from control. ^1^ AX concentration in AX-rich bread diet vs. AX concentration in white wheat flour bread diet. ^2^ Measured as glucose and insulin flux.

**Figure 2 nutrients-17-01561-f002:**
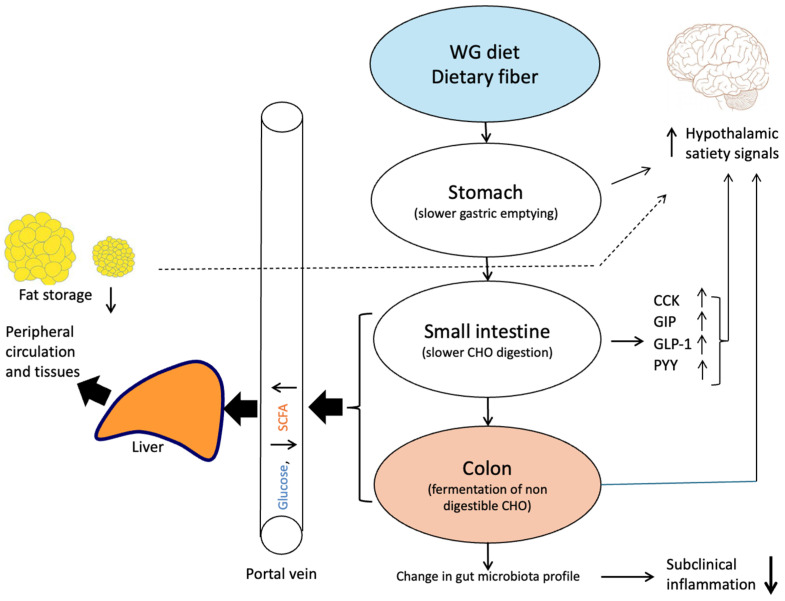
Influence of dietary fiber on the digestion and absorption processes in the gastrointestinal tract and hormonal responses. From [52]. ↑ denotes upregulation or higher concentration; ↓ denotes downregulation or lower concentration.

A dose-dependent and significantly reduced glucose response and iAUC for glucose was observed in healthy persons after the addition of 6 g or 12 g of AX-rich wheat fiber (AX-rich fraction I) to breads served as a breakfast meal [28]. The inverse relationship between the amount of AX-rich fiber in meals and the iAUC was significant (r^2^ = 0.989, *p* = 0.033). The response of the iAUC was less significant in the studies of Hartvigsen et al. [38,50], where the AX concentrate (AX-rich fraction III) was provided via bread [38] or added as a concentrate to semolina porridge alone or combined (1:1) with whole rye kernels (Table 5) [50]. Although the effect of AX on the iAUC in the individual studies of Hartvigsen et al. [38,50] was not significant, there was an overall significant negative relationship (r^2^ = 0.8267, *p* = 0.0017) between the iAUC and the dose of AX across the acute human intervention studies (Figure 3).

The variability in glucose absorption reflects differences in the rate of starch digestion, where the porous structure of the food in the controls (white wheat bread or semolina porridge) is more easily destroyed in the mouth, stomach, and small intestine than that of foods with added or intrinsic DF or AX [53]. The outcome is more rapid digestion and absorption and, consequently, a higher glycemic response in the controls than with the AX-enriched foods, despite a similar load of digestible carbohydrates. This is supported by the positive relationship between the hydrolysis rate (*k*) in vitro and the absorption rate (*k*) in vivo (Table 6) [54]. The in vitro study mimicked the rate of hydrolysis of starch in the small intestine and was based on the procedure of Englyst et al. [55], modified to include more sampling points [56]. The in vivo study was carried out on the portal vein and catheterized mesenteric artery in pigs, with a flow probe around the portal vein to monitor the blood flow rate [37]. This technique enables the absorption of glucose to be quantified [57]. The in vitro and in vivo *k* values were positively related [54], and there was a trend of lower *k* values for AXB and the other DF-enriched breads compared with the WWB control. In the in vivo study, the net portal flux increased immediately after feeding and peaked between 45 and 60 min postprandially at 9.4 mmol/min for pigs fed the WWB control and 6–7 mmol/min for the DF-rich breads. The glucose flux was significantly lower when consuming the AX-rich bread at 60 min (*p* = 0.02) than the WWB control [37]. The in vitro study further revealed that it was within the first 0–5 min that the hydrolysis of starch was significantly slower for all the DF-rich breads (7.4–10.4% starch/min) compared to the control bread (13.9% starch/min). The concentrated AX added to bread was as efficient in reducing the hydrolysis rate as the breads with intact kernels [54].

### 4.2. Medium- and Long-Term Intervention Studies

A summary of results from medium- and long-term human and animal intervention studies performed with AX concentrate from cereals is shown in Table 7 and Table 8. In contrast to acute studies, medium- and long-term intervention studies enable studies of the long-term effects of diets or ingredients on the gut microbiota, lipid parameters, and glucose homeostasis. In the study with human subjects with T2DM, it was found that the consumption of AX fiber (15 g/d) for 5 weeks significantly improved fasting glucose, and 2 h blood glucose and insulin concentrations when performing an oral glucose tolerance test (OGTT) [58]. The lower fructosamine concentration indicated an overall improved glucose homeostasis over the 5-week period. Garcia et al. [31] similarly showed decreased serum glucose, insulin, and plasma ghrelin responses when 15 g of AX was provided through breads (10 g) and as a powder (5 g) added to breakfast cereals for 6 weeks to human subjects with impaired glucose tolerance. In a parallel experiment designed as a single-blind, controlled, crossover intervention study over 6 weeks, fasting serum glucose, serum triglycerides, and apolipoprotein A-1 were significantly lower during AX consumption compared to placebo (*p* = 0.029, *p* = 0.047; *p* = 0.029, respectively), whereas no effects of AX on fasting adipokine concentrations (insulin, adiponectin, leptin, and resistin), apolipoprotein B, or the serum concentrations of unesterified fatty acids were observed [29]. It is well-known that patients with T2DM often have dyslipidemia, and the reduced serum triglyceride levels are presumably caused by the higher digesta viscosity, which reduces lipid absorption, possibly through a combination of the viscous fiber’s interaction with the micellar solubility of lipids [59] and the increased bile acid excretion caused by the viscous fiber in combination with the microbiota [44,45,60,61]. The results obtained in the two studies on glucose regulation with wheat AX concentrate [29,40] are consistent with the results of a 6-month intervention study with maize bran hemicelluloses (mostly AX). In the latter study, the maize bran hemicelluloses were provided at a dose of 10 g/d to three groups of subjects consisting of patients with impaired glucose tolerance with or without obesity and healthy non-obese controls. The long-term supplementation with the maize bran hemicellulose decreased the post-OGTT glucose and insulin curves for both groups of patients with mild T2DM, but not the healthy controls. The HbA_1c_ levels, used as a marker for glucose regulation, decreased significantly during maize bran hemicellulose supplementation in obese patients, while fasting glucose decreased in all three groups, although not significantly [62].

When combining the data on the peak glucose concentrations across the acute-, medium-, and long-term human studies, a significantly negative relationship was observed between the peak glucose level compared to run-in and the consumption of the AX-enriched diets (r^2^ = 0.58, *p* = 0.048; Figure 4). These data, together with the fructosamine and HbA1c data [29,40,62], indicate improved glucose regulation with the AX-rich diets. These aspects are supported by data from a study with Zucker diabetic fatty (ZDF) rats and rats made diabetic with streptozotocin (STZ) [44,63]. The ZDF rats were fed pelleted diets prepared from the breads that are presented in Table 2 and were used in the studies with human subjects [38] and animals [37,64], whereas the STZ-fed rats were fed AX fiber isolated from barley along with eight other DF sources. In the study with the ZDF rats, the fasting blood glucose and glucose responses to OGTT (gavage) were significantly improved for the bread enriched in AX concentrate compared to the low-DF wheat bread control and similar to what was obtained with whole grain rye breads, intrinsically rich in AX [63]. The AX-enriched bread and the whole grain rye breads also led to higher insulin sensitivity, higher insulin responses, and delayed onset of diabetes compared to the control diet low in DF. It was also found that the bread with added AX from concentrate had similar beneficial effects to the AX present intrinsically in rye breads in delaying the development of T2DM in ZDF rats. The diet intervention considerably changed gene expression in insulin-sensitive tissues [63]. The study with STZ rats also showed reduced fasting blood glucose with AX fiber, increased GLP-1, and increased microbial propionate production, but no improvement in the impaired glucose tolerance caused by OGTT [44]. In contrast to in these studies, an AX-rich diet had no effect on the glycemic response in normoglycemic pigs after a feed challenge or OGTT [65]. Pigs fed the combination of AX and β-glucan, however, had a reduced GIP response and delayed insulin peak following the feed challenge.

**Table 7 nutrients-17-01561-t007:** Medium- to long-term human and animal intervention studies on the effects of arabinoxylan on glucose homeostasis.

AX Source	AX Dose and Control	Species and Duration	Parameters Studied	Observed Effect	Reference
AX extracted from wheat co-products after starch and gluten extraction	15 g of AX-rich fiber provided through bread and muffins compared to a whole grain white flour (1:1)	Human subjects with type 2 diabetes. Intervention period: 5 weeks	Plasma glucose, insulin, fructosamine, blood lipids, blood pressure, OGTT, and fecal output	AX consumption resulted in significantly lower fasting and 2 h plasma glucose, lower 2 h insulin and serum fructosamine, and increased fecal output	[58]
AX concentrated from process water after wheat starch extraction	15 g of AX provided in breads and as powder compared to white wheat bread	Human subjects with impaired glucose tolerance. Intervention period: 6 weeks	Serum glucose, insulin, and triglycerides, and plasma total and acetylated ghrelin	AX consumption resulted in lower postprandial responses in serum glucose, insulin, and triglycerides. Compared to the placebo, total plasma ghrelin was also reduced, but acetylated ghrelin was not	[29,31]
AX concentrate from wheat by-products after gluten extraction	AX (10%) added to a wheat starch-based diet alone or combined with β-glucan (5% AX+5% β-glucan) and compared with a low-DF wheat starch diet, a wheat starch diet with added β-glucan (10%), or a whole wheat flour diet	Male normoglycemic pigs. Intervention period: 3 weeks	Plasma glucose, insulin, NEFA, GIP, GLP-1, PYY, ghrelin, glucagon, cortisol concentrations, and OGTT	AX had no effect on glycemic response following the feed challenge or the oral glucose tolerance test as determined by the area under the curve. A biphasic glucose and insulin response was detected for all pigs following the OGTT. Pigs fed the combination of AX and β-glucan had a reduced GIP response and delayed insulin peak following the feed challenge. Incretin (GLP-1 and GIP) secretion appeared asynchronous, reflecting their different enteroendocrine cell locations and responses to nutrient absorption	[65]
AX concentrate from the soluble fraction after wheat starch and gluten extraction	Pelleted diets prepared from ground and dried AX bread, white wheat bread, whole grain rye bread, whole grain rye bread with kernels, and β-glucan-rich bread	Zucker diabetic fatty (ZDF) rats. Intervention period: 7 weeks	Plasma glucose, insulin, OGTT, glucagon, triglycerides, cholesterol, HDL cholesterol, free fatty acids, HbA1c, and key genes related to insulin signaling cascade, glucose metabolism, and inflammation	AX added to wheat bread had similar beneficial effects on glycemic control according to the oral glucose tolerance test and in changing the expression of key adipose and hepatic genes to AX-rich rye breads without and with kernels	[63]

AX, arabinoxylan; NEFA, non-esterified fatty acids; GLP-1, glucagon-like peptide-1; GIP, glucose-dependent insulinotropic peptide; PYY, peptide tyrosine tyrosine.

**Table 8 nutrients-17-01561-t008:** Main effects of inclusion of arabinoxylan relative to control in diets in medium- and long-term human and animal intervention studies on parameters related to glucose homeostasis.

	AX		Concentration						
	g/d or % of DM	OGTT	Glucose	Insulin	Fructosamine/HbA1c	GLP-1	GIP	Ghrelin	Ref.
AX extracted from wheat co-products after starch and gluten extraction	15 g	↓	↓	↓	↓				[58]
AX concentrated from process water after wheat starch extraction	15 g		↓	↓				→	[29,31]
AX concentrate from wheat by-products after gluten extraction	10% vs. 0%	→	→	→		→	→	→	[65]
AX concentrate from the soluble fraction after wheat starch and gluten extraction	7.1% vs. 2.2%	↓	↓	↓	↓				[63]

AX, arabinoxylan; OGTT, oral glucose tolerance test; HbA1c, glycated hemoglobin A1c; GLP-1, glucagon-like peptide-1; GIP, glucose-dependent insulinotropic peptide. ↓ significantly reduced compared to control (*p* < 0.05)); → not significantly different from control.

## 5. Influence of AX on Insulin and Incretins

A consequence of the lifestyle common in affluent societies is an increase in body weight, which over a long time may lead to metabolic syndrome and T2DM; the latter is caused by a relative insulin deficiency due to pancreatic β-cell dysfunction and insulin resistance in multiple organs [1,66]. Based on the data presented in Table 6 and Table 8, the direction of the change in the insulin concentration and iAUC followed the same path as for glucose in the acute studies and for the glucose concentration in the medium- and long-term intervention studies. Overall, a lower rate of glucose absorption is followed by lower insulin secretion and a lower iAUC for insulin. One exception, however, was the acute human study performed by Hartvigsen et al. [38], where the glucose concentration was lower but the insulin concentration was higher after the AX-enriched bread was consumed. The same was observed in the study with catheterized pigs fed the same bread types, where the cumulated insulin secretion reached almost the same level after 240 min as for the white wheat control bread [67]. The most likely explanation for this discrepancy is the high level of associated protein in the AX-rich fraction III (Table 1); the protein concentration of the AX-rich fraction III is two to almost four times higher than that of the other AX-rich fractions.

Nutrients known to stimulate incretin secretion are glucose, fat, and protein, whereas the influence of DF is unclear [7,8]. The human studies performed with AX concentrates do not provide further clues on the effect of DF on these hormones, as there was no significant effect of AX and thereby DF intake on the secretion of GLP-1. Similarly, a diet rich in AX from whole foods and resistant starch compared with a diet rich in refined carbohydrates did not show any significant effect on GLP-1 secretion [68]. In the study of Hartvigsen et al. [50], however, there was a significant effect of adding concentrated AX to semolina porridge on the GLP-1 response. In this study, intending also to study the possible second meal effect of the diet, there was a positive correlation between plasma propionate, derived from microbial fermentation in the large intestine, and the GLP-1 concentration. The data from Hartvigsen et al. [50] corroborated findings in STZ diabetic rats and C57BL/6L male mice, where AX from barley has been found to stimulate GLP-1 secretion via G protein-coupled receptors 43 [69], primarily due to increased SCFA production [12,60]. In the second meal period, 240–360 min, there were no responses to diet of glucose, insulin, free fatty acids, GLP-1, and ghrelin [50].

## 6. Influence of AX on the Plasma Metabolome

Blood plasma samples from the acute intervention study with human subjects with metabolic syndrome [38] and the acute intervention study with catheterized pigs [37] were subjected to LC MS/MS-based metabolomics, enabling analyses of a wide array of metabolites that may be influenced by the dietary intervention [64]. Overall, the postprandial responses were similar in pigs and humans, as 21 out of 26 identified metabolites were qualitatively similar in response with the test breads despite the different basal metabolome concentrations in the plasma of the two species [64] (Figure 5). Among the similarly responding metabolites was tryptophan. Tryptophan is insulinotropic in humans [38] and could be causative for the cumulative increase in insulin secretion in the late absorption phase due to its presence in the protein of the AX concentrate [37]. The metabolites that differed between species were phosphatidylcholines, oleic acid, and carnitine. These were higher in human plasma than in plasma from pigs, possibly reflecting a higher intake of meats and fats in human subjects than in pigs. The pigs, on the other hand, had higher plasma concentrations of betaine, choline, creatinine, tryptophan, and phenylalanine, reflecting a higher dose of bread provided to the pigs (per kg bodyweight) and/or that the pigs were still growing [64]. Another interesting difference was that the metabolome of the human subjects with metabolic syndrome was more variable than that of pigs, probably reflecting a greater diversity in human subjects than in pigs (Figure 5). Nevertheless, the metabolomes in the two species clustered similarly despite differences in their basal metabolism.

Beyond the metabolites influenced by AX in the acute phase, there are numerous metabolites that changed when supplementing AX (from psyllium husks) to healthy subjects over a 3-week period, during which the AX dose was gradually increased from 5 to 15 g/d over three weeks [45]. AX supplementation had a significant and dose-dependent impact on low-density lipoproteins, and an increase in bile acids contributed to cholesterol reduction. These effects were more pronounced than when intervening with the same doses of long-chain inulin or a mix of inulin and AX. Hyperlipidemia- and hypercholesterolemia-related lipids and cholesterol metabolism typically occur in the onset and progression of diabetes. It can be noted that, while diabetes-associated dyslipidemia results in the accumulation of incompletely oxidized lipid species, leading to impaired insulin action and glucose homeostasis, AX supplementation (concentrated from *Plantago asiatica* L.) decreased the concentrations of some metabolites (12α-hydroxylated bile acids, carnitines, free-fatty acids, and lysophophatidylcholine) associated with improved bile acid and lipid metabolism. These aspects might be involved in improving hypercholesterolemia and hyperlipidemia [60,70]. Other metabolomic studies with AX supplementation in type 2 diabetic mice and high-fat-diet-induced obese mice have shown changes in metabolomic profiles regarding the cecum content [71] and liver metabolism, as indicated by liver function indicators, histopathological damage, and differential metabolites of pantothenate and CoA biosynthesis, as well as purine metabolism [35]

## 7. Influence of AX in Diets on Satiety and Plasma Ghrelin

It is generally accepted that DF consumption can reduce feelings of hunger, increase feelings of fullness, and promote satiety [72]. This suggests associations with the complex network of gut hormones regulating energy metabolism. Among these hormones, ghrelin, a peptide mainly produced and excreted in the stomach, is considered important. Ghrelin is found in the plasma in two major forms, acylated and desacylated ghrelin, which have antagonistic effects on the endocrine regulation of energy balance [73,74]. The observed effects of concentrated AX from wheat on ghrelin, however, are variable, showing promoting [75], inhibitory [29], or no effects on total ghrelin [38,50]. The same was the case when assessing the influence of AX on appetite sensation (satiety, hunger, and fullness) using the visual analog scale methodology [76]. For instance, subjects provided the AX-rich breads were better satiated and full and felt less hungry than the subjects given the white wheat bread, but only with a tendency for a reduced energy intake at the end of the study [38]. In the study where AX concentrate was added to semolina porridge, there was no effect on the feeling of hunger compared to that with the control semolina porridge, whereas concentrated AX combined with whole rye kernels reduced the feeling of hunger without influencing ghrelin [50]. The effect of concentrated AX together with rye kernels on the feeling of hunger could potentially be caused by the combined effect of soluble AX from the concentrate influencing the viscosity in the small intestine and the stimulatory effect of DF on acetate and butyrate production in the large intestine [38]. These factors can modify gastric emptying and potentially elevate the appetite-suppressing peptide tyrosine–tyrosine [7].

In addition to the dietary influence on ghrelin, there may also be a more long-term effect on appetite and energy intake modulated through the microbial metabolites, primarily SCFAs, which, through the G protein-coupled receptor 43, trigger the incretin hormone GLP-1 [69]. In a study where fermentable corn bran AX was compared with microbial non-accessible microcrystalline cellulose at a high dose (female, 25 g/d; male, 35 g/d) provided to obese subjects for a 6-week period, AX enhanced satiety after a meal and improved insulin resistance, an effect not observed with microcrystalline cellulose [77]. Using machine learning models, it was further determined that the effects on satiety could be predicted by fecal bacterial taxa that utilized AX, whereas it could not be related to the products of microbial fermentation, SCFAs. In a study, a 50/50 blend of inulin and AX was provided at a dose of 8 g/d for 3 weeks and compared to a maltodextrin control [78]. The perceived satiety and appetite were not affected by the intervention, whereas the energy intake was reduced in an ad libitum meal. Increased SCFA concentrations and fecal cell counts of Bifidobacteria and Lactobacilli were also identified.

## 8. Conclusions

Concentrates of AX from wheat with high solubility and relatively high M_w_ can be produced from co-products from the starch and gluten industries and used as ingredients in breads and breakfast cereals. Human and animal studies have shown that AX influences the viscosity of digesta in the small intestine but not the digestibility of starch. The rate and extent of glucose absorption were reduced in a fashion related to the intake of AX. Insulin, but not incretins, generally followed the same pattern as plasma glucose. Markers for glucose homeostasis (fructosamine and HbA1c) were improved, whereas the influence of AX on appetite sensation was more variable and possibly related to the microbial fermentation and production of SCFAs in the large intestine.

## 9. Future Perspectives

The current review points to co-products from the starch and gluten industries as a potential valuable resource for producing concentrated AX for use as functional and nutritional ingredients for humans with impaired glucose and lipid metabolism. These co-products are readily available worldwide and can be concentrated by various means. An element, however, that warrants consideration is protein contamination, which was high in some of the produced AX-rich fractions, impacting the obtained results.

## Figures and Tables

**Figure 1 nutrients-17-01561-f001:**
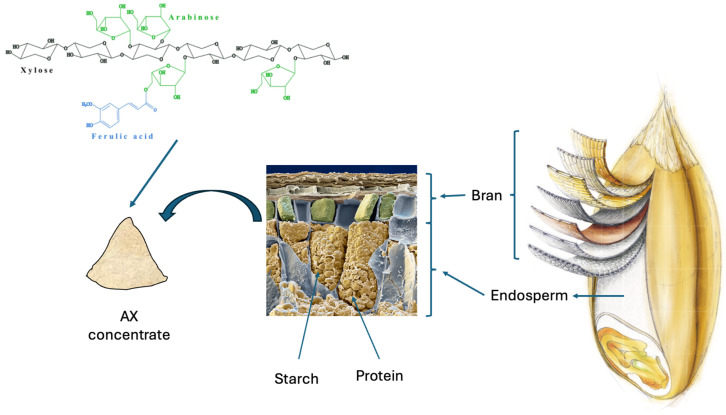
Arabinoxylan is a polymer consisting of a xylose backbone with arabinoses and ferulic acid substituents and present in all the cell walls of cereals (i.e., wheat). The molecular structure and solubility differ between cell walls within the grains and between cereal species. Adapted from Surget and Barron [26] and Eye of Science/Science Photo Library.

**Figure 3 nutrients-17-01561-f003:**
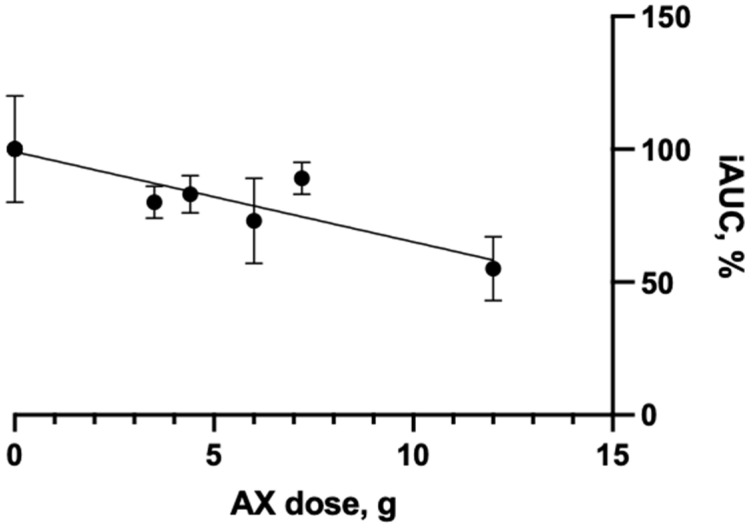
The relationship between the AX dose and iAUC obtained in acute human intervention studies. Data from studies of [28,38,50].

**Figure 4 nutrients-17-01561-f004:**
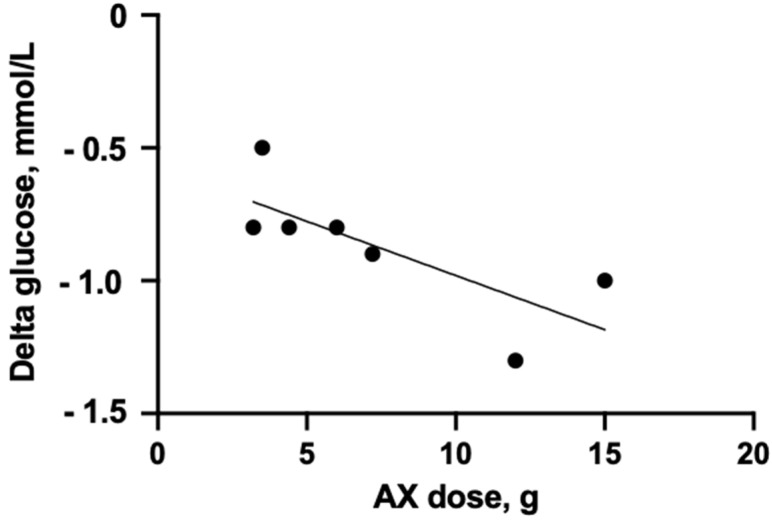
The influence of AX dose on changes in blood glucose concentration in blood from acute-, medium- and long-term human intervention studies. Data from studies of [28,31,38,50,51].

**Figure 5 nutrients-17-01561-f005:**
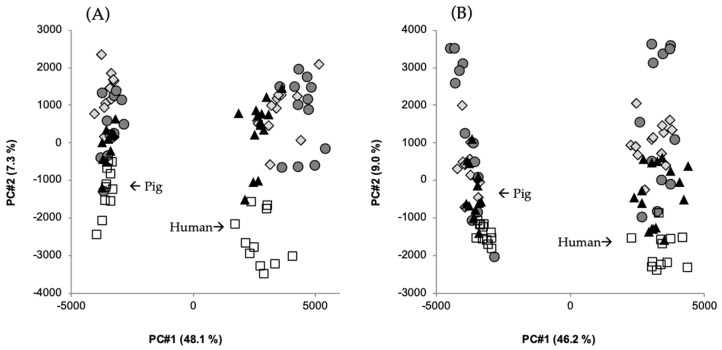
Score plots from principal component analysis of metabolic features from plasma samples from humans (N = 15) and pigs (N = 6, arterial and portal vein) given a meal of one of four different test breads with contrasting contents and compositions of DF. (**A**) 30 min postprandial and (**B**) 120 min postprandial. 
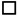
, white wheat bread (WF); 
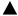
, rye bread with kernels (RK); 
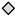
, arabinoxylan bread (AX); 
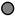
, β-glucan bread (BG). Amount of total variation accounted for by the principal components is shown in brackets. Data from [64].

**Table 2 nutrients-17-01561-t002:** Chemical compositions (g/kg dry matter (DM)) of the experimental test breads; see Christensen et al. [37] for details.

Chemical Composition	WWB	WRB	WRBK	AXB	BGB
Dry matter (g/kg as is)	634	520	543	708	615
Total starch	711	588	608	514	612
Total sugars	47	32	23	39	58
Dietary fiber					
LMW non-digestible carbohydrates	30	52	53	81	10
Resistant starch	4	9	14	7	18
Total NSP (soluble NSP)	35 (17)	134 (53)	139 (50)	116 (86)	163 (54)
Cellulose	6	19	18	6	53
β-Glucan (soluble β-glucan)	3 (2)	21 (7)	19 (4)	3 (2)	52 (40)
AX (soluble AX)	17 (13)	76 (36)	77 (37)	78 (66)	32 (9)
Total non-digestible carbohydrates ^a^	69	195	206	204	190
Klason lignin	8	14	14	8	9
Total dietary fiber ^b^	77	209	220	212	199

WWB, white wheat bread; WRB, whole grain rye bread; WRBK, whole grain rye bread with kernels; AXB, wheat bread with arabinoxylan concentrate; BGB, wheat bread with β-glucan concentrate; ^a^ Calculated as total non-starch polysaccharides (NSP) + low-molecular-weight (LMW) non-digestible carbohydrates + resistant starch. ^b^ Calculated as Total NSP + LMW non-digestible carbohydrates + resistant starch + lignin.

**Table 3 nutrients-17-01561-t003:** Digestibility of starch, AX, and NSP; concentrations of AX and NSP; and viscosity of extracts of ileal digesta when feeding bread diets; see [30,32] for details.

	Diet					
	WWB	WRB	WRBK	AXB	BGB	SE
Starch, %	99 ^a^	98 ^b^	97 ^c^	99 ^a^	96 ^c^	0.2
AX, %	28	27	32	11	31	6.7
NSP, %	17 ^b,c^	27 ^a,b,c^	29 ^a,b^	13 ^c^	38 ^a^	5.4
AX, g/kg DM	108 ^c^	212 ^b^	190 ^b^	278 ^a^	95 ^c^	8.9
NSP, g/kg DM	256 ^d^	375 ^bc^	354 ^c^	416 ^a,b^	424 ^a^	13.6
Viscosity (mean and 95% CI, mPa.S)	5.9 (3.3–10.5) ^b^	8.4 (4.7–15.0) ^b^	7.4 (4.2–13.3) ^b^	15.5 (8.6–27.6) ^a^	2.6 (1.5–4.7) ^c^	

WWB, white wheat bread; WRB, whole grain rye bread; WRBK, whole grain rye bread with kernels; AXB, wheat bread with arabinoxylan concentrate; BGB, wheat bread with β-glucan concentrate; SE, standard error; CI, confidence interval; ^a,b,c,d^ values with different superscript letters are significantly different.

**Table 6 nutrients-17-01561-t006:** In vitro and in vivo data for the digestion and absorption rates (*k*), asymptote, inflection point, and incremental area under the curve (iAUC) of breads varying in dietary fiber; see [37,54] for details.

	Breads						
	WWB	WRB	WRBK	AXB	BGB	SEM	*p*-Value
In vitro							
*k*, % hydr./min	0.1595 ^a^	0.1462 ^ab^	0.1048 ^ab^	0.1000 ^ab^	0.0744 ^b^	0.02	0.06
Asymptote, %	89.8	89.6	89.5	94.4	90.2	3.1	0.75
In vivo							
*k*, % absorption/min	0.0227 ^a^	0.0195 ^a^	0.0210 ^a^	0.0191 ^a^	0.0167 ^a^	0.023	0.085
Inflection point	55.4	66.0	66.0	60.6	59.8	7.3	0.82
Asymptote, %	71.8	56.8	62.6	68.5	72.2	10.4	0.85
*k*_in vitro_/*k*_in vivo_	7.02	7.50	4.99	5.23	4.45		
iAUC	100	70	83	74	89	12	0.10

WWB, white wheat bread; WRB, whole grain rye bread; WRBK, whole grain rye bread with kernels; AXB, wheat bread with arabinoxylan concentrate; BGB, wheat bread with β-glucan concentrate. ^a,b^ Mean values with different superscript letters are significantly different. iAUC, incremental area under the curve.

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
