# Peer review of "Arabinoxylan Concentrate from Wheat as a Functional Food Ingredient to Improve Glucose Homeostasis"

_nutrients, 2025, doi:10.3390/nu17091561_

Round 1

Reviewer 1 Report

Comments and Suggestions for Authors

This manuscript summarized the impact of AX as a concentrated food ingredient on digestion and absorption of glucose and glucose homeostasis. However, This review lacks novelty; the existing literature already includes many similar articles, and this manuscript does not provide any new insights to the field. Additionally, the manuscript does not provide a comprehensive, critical, and deep discussion of this topic. As a review article, the discussion is superficial, lacking the necessary depth and breadth of knowledge required to make a meaningful contribution to the field. In addition, the references throughout the manuscript require updating, as nearly all cited works are from five or more years ago, with some dating back ten years. This indicates that the authors have not conducted a thorough examination and analysis of the current state of research in their field. For the manuscript to be considered for publication, the authors must provide a more in-depth discussion and analysis, citing relevant literature from the past five years.

(1) Are there any studies on the application of AX in other food products? This research only discusses its application in bread.

(2) Figure 6 is unclear; please provide a clearer image.

(3) The literature cited in Table 7 is insufficient, with the majority of references being published over five years ago. This does not adequately reflect the latest research developments regarding medium to long-term supplementation of AX in both humans and animals.

(4) Line 384: The analysis of AX's effects on insulin and gut hormones is overly simplistic and lacks depth. While AX is known to influence insulin sensitivity, insulin secretion, and GLP-1 secretion, this manuscript does not delve into these aspects comprehensively.

(5) Line 407: The discussion on the impact of AX on plasma metabolomics is minimal, providing only a superficial examination of its effects on basic metabolites such as amino acids. Research has shown that AX affects cholesterol levels in plasma, lipoprotein distribution, various inflammatory markers, and oxidative stress status. However, this manuscript only briefly touches upon a few basic metabolites. Additionally, this section only cited three literatures, and all of them published over ten years ago.

(6) Line 436: The analysis of AX's effects on satiety and plasma ghrelin levels is too superficial, and the cited literature is predominantly from five years ago. This suggests that the authors have not explored the latest publications in this field and lack a comprehensive understanding of AX.

Author Response

This manuscript summarized the impact of AX as a concentrated food ingredient on digestion and absorption of glucose and glucose homeostasis. However, This review lacks novelty; the existing literature already includes many similar articles, and this manuscript does not provide any new insights to the field. Additionally, the manuscript does not provide a comprehensive, critical, and deep discussion of this topic. As a review article, the discussion is superficial, lacking the necessary depth and breadth of knowledge required to make a meaningful contribution to the field. In addition, the references throughout the manuscript require updating, as nearly all cited works are from five or more years ago, with some dating back ten years. This indicates that the authors have not conducted a thorough examination and analysis of the current state of research in their field. For the manuscript to be considered for publication, the authors must provide a more in-depth discussion and analysis, citing relevant literature from the past five years.

R: I don’t know the lacking litterature that have been published but not included in the review except a study with mice (Luo, D.; Li, X.; Geng, M.; Zhang, Y.; Lan, H.; Li, J.; Qi, C.; Bai, Z.; Huang, J. Effect of Arabinoxylan from Wastewater Generated during Vital Wheat Gluten Production on Liver Metabolism in Type 2 Diabetic Mice. Foods 2023, 12, doi:10.3390/foods12142640), that, however, was not well suited to be put into Table 7 and 8 as it does not provide glucose and insulin values. The review is foced on AX concentrate and not AX as intact component present as soluble and insoluble components in natural foods. When searching the litterature we were surprised to find only few new papers in this field. The aims of the review have been made clearer by modifying the title to: “Arabinoxylan concentrate from wheat as a functional ingredient to improve glucose homeostasis” and revising the abstract. Regarding the other points raised we have tried to deepen the discussion and including more litterature.

(1) Are there any studies on the application of AX in other food products? This research only discusses its application in bread.

R: To our knowledge this is not the case.

(2) Figure 6 is unclear; please provide a clearer image.

R: Done.

(3) The literature cited in Table 7 is insufficient, with the majority of references being published over five years ago. This does not adequately reflect the latest research developments regarding medium to long-term supplementation of AX in both humans and animals.

R: This review is on AX concentrates from cereals and not on AX present intrinsic as soluble and insoluble components in the cell walls or on other AX sources. As far as we can see the litterature summarised in Table 7 is what is available on the medium- and long-term studies with AX concentrates from cereals. The text, however, have been expanded by including a human study using AX from psyllium husk and rat and mice studies using AX from Plantago asiatica L.

(4) Line 384: The analysis of AX's effects on insulin and gut hormones is overly simplistic and lacks depth. While AX is known to influence insulin sensitivity, insulin secretion, and GLP-1 secretion, this manuscript does not delve into these aspects comprehensively.

R: The text have been modified and complemented with additional litterature.

(5) Line 407: The discussion on the impact of AX on plasma metabolomics is minimal, providing only a superficial examination of its effects on basic metabolites such as amino acids. Research has shown that AX affects cholesterol levels in plasma, lipoprotein distribution, various inflammatory markers, and oxidative stress status. However, this manuscript only briefly touches upon a few basic metabolites. Additionally, this section only cited three literatures, and all of them published over ten years ago.

R: The section has been expanded to include other studies performed with AX from other sources than cereals.

(6) Line 436: The analysis of AX's effects on satiety and plasma ghrelin levels is too superficial, and the cited literature is predominantly from five years ago. This suggests that the authors have not explored the latest publications in this field and lack a comprehensive understanding of AX.

R: This section has been expanded by including medium- and long-term studies performed with AX from other sources.

Reviewer 2 Report

Comments and Suggestions for Authors

Many articles have been written about the role of fiber as a food ingredient. Recently, its soluble fraction has attracted particular interest from researchers. Arabinoxylan is a naturally occurring polysaccharide that is part of the fiber consumed by humans. It consists of a basic chain made of a D-xylopyranose unit connected by a β(1,4) glycosidic bond, which chain has α(1,2) or α(1,3) branches with L-arabinofuranose residues. Its properties depend on the number of branches, which is related to its origin.

The wealth of scientific literature that has been published on arabinoxylan in recent years makes it a great challenge to choose from it the one that best illustrates its properties as a food ingredient. The authors have done very well. From among the factors resulting from the consumption of arabinoxylan described in the literature in recent years, the authors have made an interesting and logical choice and supported it with well-chosen citations. The form of presentation of the cited information is very clear, especially in the form of tables. The conclusions at the end of the article are most appropriate and are fully reflected in its content.

Author Response

Many articles have been written about the role of fiber as a food ingredient. Recently, its soluble fraction has attracted particular interest from researchers. Arabinoxylan is a naturally occurring polysaccharide that is part of the fiber consumed by humans. It consists of a basic chain made of a D-xylopyranose unit connected by a β(1,4) glycosidic bond, which chain has α(1,2) or α(1,3) branches with L-arabinofuranose residues. Its properties depend on the number of branches, which is related to its origin.

The wealth of scientific literature that has been published on arabinoxylan in recent years makes it a great challenge to choose from it the one that best illustrates its properties as a food ingredient. The authors have done very well. From among the factors resulting from the consumption of arabinoxylan described in the literature in recent years, the authors have made an interesting and logical choice and supported it with well-chosen citations. The form of presentation of the cited information is very clear, especially in the form of tables. The conclusions at the end of the article are most appropriate and are fully reflected in its content.

R: Thank you very much for the kind comments. Please also note that modifications have been made to the revised manuscript.

Reviewer 3 Report

Comments and Suggestions for Authors

Dear Authors,

Thank you for submitting your manuscript titled "Arabinoxylan as a functional food ingredient to improve glucose homeostasis" to Nutrients. I have carefully reviewed your work and would like to provide feedback to help improve the quality and comprehensiveness of your paper. Please find my detailed comments below:

Your manuscript provides valuable insights into the potential of arabinoxylan as a functional food ingredient to improve glucose homeostasis. However, one key limitation is the reliance on individual datasets or case studies (e.g., 1 or 2 examples per case). While these studies contribute valuable information, they lack the breadth required for robust conclusions.

For a more comprehensive review, I strongly recommend expanding the data set to include additional studies and conducting a meta-analysis of the available literature. This will allow you to evaluate the effects of arabinoxylan in a broader range of experimental conditions and ensure that conclusions are supported by larger and more statistically reliable datasets. Consider including studies with diverse variables such as participant age, health condition, and arabinoxylan dosages to generate broader applicability.

The studies referenced in your manuscript should be carefully reviewed to confirm whether experimental conditions or variables align across datasets. For example, differences in participant characteristics, intervention durations, or outcome measures could influence the comparability of the findings.

I suggest creating a systematic table summarizing each referenced study, detailing key characteristics and variables (e.g., sample size, intervention protocol, results, etc.). Such an approach would allow readers to more easily understand the scope and boundaries of the research.

I look forward to seeing a revised version of your manuscript that incorporates these improvements.

Author Response

Dear Authors,

Thank you for submitting your manuscript titled "Arabinoxylan as a functional food ingredient to improve glucose homeostasis" to Nutrients. I have carefully reviewed your work and would like to provide feedback to help improve the quality and comprehensiveness of your paper. Please find my detailed comments below:

Your manuscript provides valuable insights into the potential of arabinoxylan as a functional food ingredient to improve glucose homeostasis. However, one key limitation is the reliance on individual datasets or case studies (e.g., 1 or 2 examples per case). While these studies contribute valuable information, they lack the breadth required for robust conclusions.

For a more comprehensive review, I strongly recommend expanding the data set to include additional studies and conducting a meta-analysis of the available literature. This will allow you to evaluate the effects of arabinoxylan in a broader range of experimental conditions and ensure that conclusions are supported by larger and more statistically reliable datasets. Consider including studies with diverse variables such as participant age, health condition, and arabinoxylan dosages to generate broader applicability.

The studies referenced in your manuscript should be carefully reviewed to confirm whether experimental conditions or variables align across datasets. For example, differences in participant characteristics, intervention durations, or outcome measures could influence the comparability of the findings.

I suggest creating a systematic table summarizing each referenced study, detailing key characteristics and variables (e.g., sample size, intervention protocol, results, etc.). Such an approach would allow readers to more easily understand the scope and boundaries of the research.

I look forward to seeing a revised version of your manuscript that incorporates these improvements.

R: After receiving the comments from the reviewers we have become aware that the title does not in a sufficient way reflect the intention with the paper. The titles “Arabinoxylan concentrate from wheat as a functional ingredient to improve glucose homeostasis” and abstract have therefore been revised. Complementary litterature have also been included.

Reviewer 4 Report

Comments and Suggestions for Authors

The similarity index is too high. The authors should avoid this in a future submission.

The type of review should be indicated in the title, in the abstract, and the whole manuscript.

The abstract should be structured as recommended by the journal’s guidelines.

The study’s aims are not mentioned in the abstract. The methods are also not indicated (searched databases, search strategy, inclusion/exclusion criteria, searched keywords…); The conclusions are not clear, and future perspectives are missing.

The introductory section is adequate, but it is missing a Methods section. See the example of section 2 of this paper: https://www.mdpi.com/2304-8158/10/6/1175

Figures 1 and 2 have no references and no information about the copyright. Please provide this information.

I think that the section about the influence of AX in diets on satiety and plasma ghrelin is very relevant, and it should be expanded.

Include the future perspective at the end of your conclusions.

Author Response

The similarity index is too high. The authors should avoid this in a future submission.

R: When looking through similarity index report I feel that the reason to why this is concidered too high is because the there are tables that is copied and with almost the same text as in the original publications.

The type of review should be indicated in the title, in the abstract, and the whole manuscript.

The abstract should be structured as recommended by the journal’s guidelines.

R: The abstract has been structured as seen in other reviews.

The study’s aims are not mentioned in the abstract. The methods are also not indicated (searched databases, search strategy, inclusion/exclusion criteria, searched keywords…); The conclusions are not clear, and future perspectives are missing.

R: The study aim has been included in the abstract and the kind of search that have been performed has been described as the last part of the Introduction.

The introductory section is adequate, but it is missing a Methods section. See the example of section 2 of this paper: https://www.mdpi.com/2304-8158/10/6/1175

R: We hope the changes made is adequate.

Figures 1 and 2 have no references and no information about the copyright. Please provide this information.

R: This has been added.

I think that the section about the influence of AX in diets on satiety and plasma ghrelin is very relevant, and it should be expanded.

R: More information on findings for medium- and long-term studies have been added.

Include the future perspective at the end of your conclusions.

R: A section on future perspectives have been included.

Reviewer 5 Report

Comments and Suggestions for Authors

Can you explain why the arabinoxylan effects on glucose level is not so important if you analyse the data that are presented in table 5.

Arabinoxylan modify the absorption of other nutrients?

In the studies analysed in the present paper arabinoxylan is associated with bread or other type of cereal. Do you found any information about the use of arabinoxylan in association with sugar or food rich in sugar?

The information from tables and figures are well presented.

Author Response

Can you explain why the arabinoxylan effects on glucose level is not so important if you analyse the data that are presented in table 5.

R: The data in Table 5 show in general a reduction in the glucose concentration but not the iAUC for glucose. The reason for the less pronounced effect on the iAUC-glucose is probably that this paramets typically become more variable they way it is calculated.

Arabinoxylan modify the absorption of other nutrients?

R: This is a good question but I don’t know at present.

In the studies analysed in the present paper arabinoxylan is associated with bread or other type of cereal. Do you found any information about the use of arabinoxylan in association with sugar or food rich in sugar?

R: This is an interesting question but I havn’t seen data on that.

The information from tables and figures are well presented.

R: Thank you.